# Epidemiology and Characteristics of Spondyloarthritis in the Predominantly Afro-Descendant Population of Martinique, a French Caribbean Island

**DOI:** 10.3390/jcm11051299

**Published:** 2022-02-27

**Authors:** Fabienne Louis-Sidney, Valentine Kahn, Benoit Suzon, Michel De Bandt, Christophe Deligny, Serge Arfi, Georges Jean-Baptiste

**Affiliations:** 1Service de Rhumatologie, CHU de Fort de France, 97200 Fort de France, Martinique, France; valentine.kahn@chu-martinique.fr (V.K.); michel.debandt@chu-martinique.fr (M.D.B.); georges.jeanbaptiste3@gmail.com (G.J.-B.); 2Service de Médecine Interne, CHU de Fort de France, 97200 Fort de France, Martinique, France; benoit.suzon@chu-martinique.fr (B.S.); christophe.deligny@chu-martinique.fr (C.D.); serge.arfi@chu-martinique.fr (S.A.)

**Keywords:** spondyloarthritis, epidemiology, Martinique, diagnosis, HLA-B27

## Abstract

(1) Background: The prevalence of Spondyloarthritis (SpA) varies significantly in different regions and ethnic groups due several factors such as heterogeneity in study populations, the diversity of classification criteria used in epidemiological studies, the prevalence variability of HLA-B27 or disparity in healthcare access. To our knowledge, there is no data on SpA in Martinique, a French region in the Caribbean with a predominantly Afro-descendant population and a high level of healthcare. (2) Methods: This was a retrospective study of all SpA patients treated at the Fort de France University Hospital between 1 January 1997 and 1 January 2008. (3) Results: In our cohort of 86 SpA patients, age at diagnosis was late (41 years old), ankylosing spondylitis (AS) was the most frequent sub-type (60.5%), inflammatory bowel disease was the most frequent extra articular feature (23.3%) and no one had personal familial history of the disease. Inflammatory syndrome concerned 55.6% of patients, no one was positive for HIV and HLA-B27 positivity was low (42.2%). However, HLA-B27 was statistically associated with AS. Out of 64 patients, 41 had sacroiliitis. (4) Conclusion: To our knowledge, this is the first comprehensive descriptive study of SpA subtypes in Martinique, a French region in the Caribbean. We report clinical and biological similarities in our SpA cohort with those of sub-Saharan Africa and with SpA subtypes reported in Afro-descendant populations.

## 1. Introduction

Spondyloarthritis is a chronic inflammatory rheumatic disease involving different pathologies with common clinical, biological, radiographic, and genetic features. These features are incorporated in several SpA classification criteria [1] and particularly in the most recent 2009 ASAS criteria [2]. The prevalence of the disease varies significantly in different regions and ethnic groups. This phenomenon can be explained by heterogeneity in study populations [1], differences in healthcare access in each region, the diversity of classification criteria used in epidemiological studies [1] and differences in the prevalence of HLA-B27 [1,3].

To our knowledge, there is no data concerning Spondyloarthritis in Martinique. This French region of the Caribbean is very unique, as most of its population is of African descent [4] originally from Central Africa and West Africa [5]; the healthcare system is comparable to mainland France, with free access to medical care for each citizen, and a single hospital represented by the university hospital center provides care for a population under 400,000 inhabitants. There are no private clinics providing care for SpA in Martinique. Another particularity is that HLA-B27 prevalence is very low there, accounting for 2% of the population, and Martinicans share HLA allele characteristics with Africans based on the study by Bera et al. [6].

It is interesting to see that low HLA-B27 prevalence is often considered as a main contributor to low SpA prevalence in sub-Saharan Africa [7,8,9,10]. However, available data remains poor and restricted to case series or hospital cohorts in sub-Saharan Africa given the limited access to healthcare in the region [10,11,12]. SpA displays specific characteristics in black populations: low HLA-B27 prevalence and rare family history (PFH) of SpA/inflammatory bowel disease/acute anterior uveitis(AAU)/psoriatic arthritis [13,14]. Another interesting point is that clinical characteristics and the epidemiology of SpA has been changing over the last few decades in sub-Saharan Africa, with an increase in psoriatic arthritis and reactive arthritis reported among SpA cohorts, along with a high HIV/AIDS prevalence in the region [9,12,15,16,17,18]. This highlights the interaction of genetics and environment in disease phenotypes.

We aim to assess the characteristics of SpA in a population with a high level of care in Martinique.

## 2. Materials and Methods

We conducted a retrospective study of all SpA patients treated at the Fort de France University Hospital between 1 January 1997 and 1 January 2008.

Patients with a diagnosis of SpA according to “European spondylarthropathy study group” (ESSG) criteria for SpA [19] made by an internist or rheumatologist and a disease onset after 15 years of age were included. Undifferentiated spondyloarthritis (uSpA) was diagnosed if the ESSG criteria were fulfilled, but no diagnosis of ankylosing spondylitis (AS), psoriatic arthritis (PsA), reactive arthritis (ReA) or inflammatory associated spondyloarthritis (IBD SpA) could be made. Diagnoses of PsA and AS were based on CASPAR criteria [20] and New York modified criteria, respectively [21]. The diagnosis of “Synovitis, Acne, Pustulosis, Hyperostosis, Osteitis” syndrome (SAPHO) was based on the classification criteria proposed by Kahn et al. in 1994 [22].

Non-inclusion criteria were insufficient clinical and para-clinical information to fulfill ESSG criteria for SpA and a different diagnosis after reviewing the patient’s file.

Inclusion was performed from the hospital medicalized information system program (PMSI) database with the following ICD-10 codes: M02 (post-infective and reactive arthropathies), M03 (post-meningococcal arthritis), M45 (ankylosing spondylitis), M07 (enteropathic arthropathies), M08 (juvenile arthritis), M09 (juvenile arthritis in diseases classified elsewhere), K50.9 (cervical disc disorder, unspecified), M07.4 (arthropathy in Crohn’s disease), M07.5 (arthropathy in ulcerative colitis), K51 (ulcerative colitis) and L40.5 (arthropathic psoriasis).

Investigations were carried out following the rules of the Declaration of Helsinki of 1975 (revised in 2013), and our study received approval from the institutional review board (IRB) of the University Hospital of Fort de France.

### 2.1. Evaluation Criteria

Clinical and para-clinical evaluation criteria were collected retrospectively in patients’ files.

Clinical criteria were: age at inclusion, age at SpA diagnosis, family history (PFH) of SpA or psoriatic arthritis (PsA) or inflammatory bowel disease (IBD) or acute anterior uveitis (AAU), personal history urethritis or acute diarrhea, extra articular features such as IBD or AAU, time between onset of rheumatological symptoms and SpA diagnosis, characteristics of onset presentation (axial or peripheral), occurrence of peripheral signs (arthritis, arthralgia, enthesitis), occurrence of buttock pain, occurrence of inflammatory back pain, occurrence of spinal ankylosis defined by a Schober index below 10 + 4 cm and/or a thoracic ampliation of less than 2.5 cm.

Para-clinical criteria were evidence of a raised C reactive protein (CRP) over 10 mg/L, occurrence of HLA-B27 positivity, evidence of HIV infection, occurrence of radiographic or magnetic resonance (MRI) sacroiliitis. Radiographic sacroiliac images were taken from the medical records and were reviewed according to the New York criteria of 1966. If normal, patients’ files were reviewed to see whether or not a positive MRI sacroiliitis result had been mentioned.

Treatments received since the onset of the disease were also investigated: use of non-steroidal anti-inflammatory drugs (NSAIDs), steroids, conventional disease-modifying anti-rheumatic drugs such as methotrexate (MTX), leflunomide (LEF) or salazopyrin (SLZ) and use of TNF inhibitors.

### 2.2. Statistical Analysis

Qualitative data are described as number and percentage N (%) and quantitative data as mean (SD), as appropriate. Chi-square test was used to identify any statistically significant differences in frequencies between SpA subtypes. A Student’s *t*-test was used to identify any statistically significant differences in means between SpA subtypes. *p*-value was fixed at 0.05.

## 3. Results

Between January 1997 and January 2008, 146 patients were identified through the hospital database: 71 and 75 patients from rheumatology and internal medicine departments, respectively. The flow chart is shown in Figure 1.

Eighty-six patients met ESSG criteria for SpA. Demographic and clinical characteristics are presented in Table 1. The male:female ratio was 1.3. Peripheral features were predominant at disease onset, but axial symptoms tended to increase and become more frequent at follow up.

Ankylosing spondylitis was the most frequent subtype, with 34 (65.4%) and 18 (34.6%) patients satisfying the modified New York criteria for definite AS and probable AS, respectively. Spinal enthesis ossification was described in 11 AS patients (21.2%) and 82.9% had radiographical sacroiliitis (Table 2). No SAPHO was reported. IBD was the most frequent extra articular feature in this SpA cohort (23.3%). Inflammatory syndrome concerned 55.6% of patients, HLAB27 positivity 42.2% of patients and no one was HIV positive. Compared to all other SpA subtypes, back pain, HLAB27 and sacroiliitis were significantly more frequent in AS, respectively; *p* = 0.029, *p* < 0.001 and *p* = 0.0027 (Table 2).

SpA treatments are described in Table 3. Conventional DMARDs were mostly used in our cohort. AS and IBD SpA patients were more likely to receive TNF inhibitors.

## 4. Discussion

Between 1997 and 2008, we reported 86 SpA cases in the University Hospital of Martinique. To our knowledge, this is the first descriptive study of SpA subtypes in an Afro-descendant population. Only one previous study of this type has been conducted in a sub-Saharan African population [13]. The minimal prevalence of the disease in Martinique was 2.3/10,000 inhabitants for a population size of 368,783 in 2008 [23]. The estimated prevalence is low compared to European countries, where it ranges from 30/10,000 to 190/10,000 [24,25,26,27], and North America, where it has been estimated at 135/10,000 [28] based on a 2016 meta-analysis. The 1:1 male-to-female ratio in our cohort is similar to that reported in mainland France [24]. The mean age at diagnosis is apparently late, particularly in AS, where it is 42 years, compared to less than 35 years in ASAS, DESIR and SPACE axial spondyloarthritis cohorts, which are mostly Caucasian [29,30,31]. These data are consistent with the literature on sub-Saharan African and Afro-descendant AS cohorts [8,9,10]. None of our patients had PFH. This finding is consistent with those in sub-Saharan African and Afro-descendant SpA populations, where no or little PFH is reported [9,13,14,32]. In Caucasian and Asian populations, PFH appears to be more common and is part of the ASAS and ESSG diagnostic criteria.

In DESIR, SPACE and ASAS cohorts, mostly Caucasian and Asian, PFH is associated with HLAB27 positivity and thus with possible excess risk of axial spondyloarthritis 22,23. [29,30]. One additional finding in our SpA cohort is a low prevalence of HLAB27, especially in the AS subtype, with a 54.3% prevalence estimate.

Such low prevalence is comparable to that reported in sub-Saharan African and Afro-descendant populations in the Americas. HLA-B27 was found in 56% of AS in a Burkinabe cohort and between 40 and 62.5% of Afro-American AS. These findings are in stark contrast with those of the mainly Caucasian European AS cohorts [33,34,35] and those of white American AS cohorts [3,36,37]. This low prevalence in sub-Saharan African AS is often attributed to the scarcity of HLA-B27 in these populations [38]. Martinique has the special characteristic of being inhabited by a largely Afro-descendant population [4] from Central and West Africa [5]. In 2001, Bera et al. reported the paucity of the HLA-B27 allele in the Martinican population, which partly explains the rarity of AS in Martinique, as well as numerous HLA profiles shared with Sub-Saharan African populations [6]. Yet, this can only partially be explained by the genetics. A study of the Fula population in Gambia [39] reported a 6% prevalence of HLA-B27, among which, no AS was observed. These data suggest that there may be some protective genetic or environmental factors in African populations that have yet to be identified.

Low PFH and HLA-B27 rates in both our cohort and in sub-Saharan African SpA suggest that SpA classification criteria, including the 2009 ASAS criteria, might be less effective in detecting the disease in these population groups.

IBD is highly preponderant among extra articular presentations, and IBD-associated SpA is the second most frequent subtype in our study cohort. Very few data are available on IBD-associated SpA in sub-Saharan African and afro-descendant populations. A unique study conducted on a Congolese hospital cohort of SpA reports the very low prevalence of IBD-associated SpA with a 0.1% estimation [13]. In addition, IBD prevalence is also reported as being lower in African Americans compared to Caucasians [40,41] and lower in the French West Indies compared to mainland France, according to Edouard et al. [42]. If IBD is seemingly low in sub-Saharan Africa, it is hard to affirm given the small number of studies on the subject and the inherent difficulties of access to healthcare in the region [43,44,45]. The high reported prevalence of IBD in our cohort is multi-factorial: this is a hospital cohort, where supposedly more severe patients are attended to and the majority of IBD cases are attended to in the only hospital on the island, where (1) most gastroenterologists are practicing, and (2) biotherapies are exclusively available. Moreover, IBD-associated SpA is much more frequent than PsA in our cohort, whereas PsA is, along with AS, one of the most common subtypes of PsA in European SpA cohorts [24,25,46]. This lower rate of PsA is also reported in African-descendant and sub-Saharan African populations. Kerr et al. report a significantly lower PsA prevalence in African Americans compared with Caucasians [47]. However, Afro-Caribbeans, the predominant group of African descent in a London PsA cohort, account for 11.6% of patients compared to 53.4% for Caucasians [14]. Afro-Caribbeans constituted 1.1% of the population of England and Wales in 2011 compared with 80.5% of white British [48]. Despite the lower proportion of Afro-Caribbean PsA than that of Caucasians, it is worth noting the latter’s overrepresentation in PsA compared to national census data. Two reviews of the literature report a low prevalence of PsA in sub-Saharan African populations with an increase in case numbers coinciding with the HIV epidemics in the region [9,17]. None of our 86 patients were HIV positive, whereas the incidence of AIDS in Martinique was higher than in mainland France at the same time [49]. This suggests that other environmental and genetic factors might play a protective role in the emergence of the disease phenotype.

The clinical presentation of SpA in our population is unusual considering the preponderance of peripheral symptoms (72.3% of SpA), including in axial SpA (63.5% of AS). On the other hand, López-Medina et al. reported peripheral symptoms in only 30.2% of axial SpA (Ax SpA) in a mostly Caucasian and Asian cohort [50]. These findings are comparable to those of Lebughe et al., where the estimated prevalence of peripheral involvement in a cohort of Congolese axSpA is 52.9% [13].

Our study has several limitations. It is retrospective with several missing data, particularly concerning sacroilliac imaging and HLA-B27, as mentioned in Table 1 and Table 2. The prevalence of the disease is probably underestimated, since it is a hospital-only cohort. However, only 5 out of 12 rheumatologists on the island practiced outside of the hospital and referred their patients to the only hospital rheumatology department on the island.

The high proportion of patients receiving anti-TNF drugs suggests that we report more severe thus more symptomatic SpA cases, which might not be representative of SpA patients followed outside of the hospital.

At the time of the study, French legislation did not authorize research based on ethnicity. We could not determine ethnicity for each patient but based on (1) data from the literature on Martinicans’ HLA profile, (2) results from study populations in the Caribbean and (3) similarities between our study cohort and others of Sub-Saharan African origin, we can assume that our cohort was mostly of African descent.

## 5. Conclusions

To our knowledge, this is the first comprehensive descriptive study of SpA subtypes in Martinique, a French island in the Caribbean. We report clinical and biological similarities in our SpA cohort with those of sub-Saharan African countries and SpA subtypes in Afro-descendent patients [14,47]. This preliminary study offers interesting perspectives for clinical research. In fact, SpA is a complex group of diseases in which genetics and the environment play an important role. Similarities but also differences between populations with different ethnic or geographical ancestry are useful to aid the understanding of SpA pathophysiology in the era of personalized medicine. The growing use of MRI for the study of sacroiliac joints over the last decade and the development of the new ASAS criteria for AxSpA in 2009 [2] have led to earlier and more consistent diagnoses of the condition. A new investigation on the prevalence of radiographic and non-radiographic AxSpA in the Martinican population is in progress, which started in 2009.

## Figures and Tables

**Figure 1 jcm-11-01299-f001:**
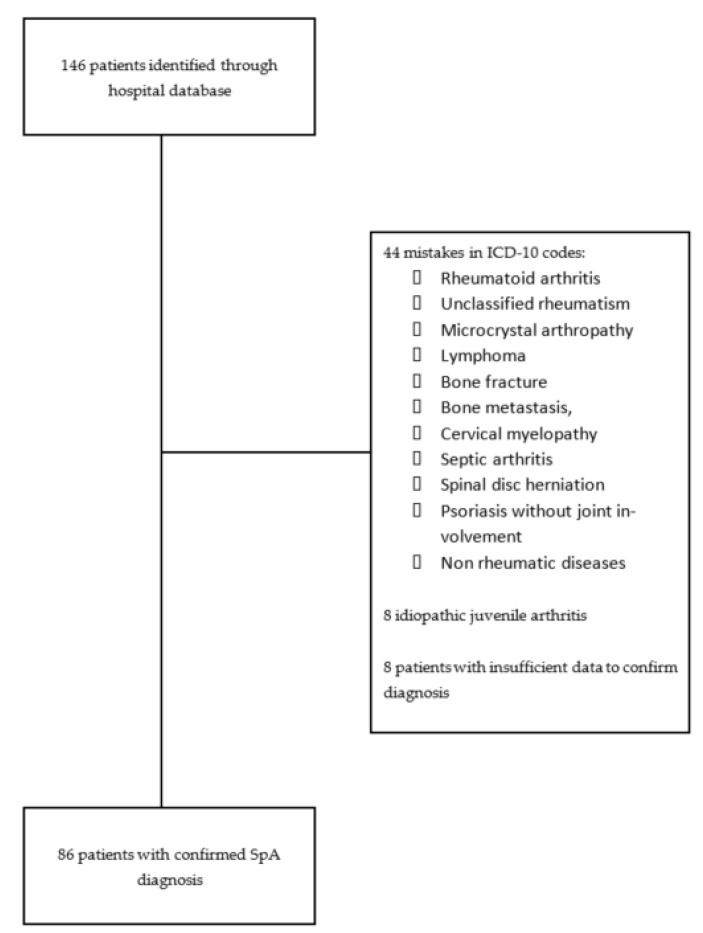
Flow chart describing Martinican SpA recruitment.

**Table 1 jcm-11-01299-t001:** Clinical, biological and radiologic characteristics of Martinican SpA patients.

Characteristics of Study Population		Total Patients (N)
Male—N (%)	48 (55.8)	86
Age at diagnosis—mean ± SD (min–max)	41 ± 14.6 (18–79)	86
Duration of disease—mean ± SD (min–max)	11 ± 5.3 (1–29)	86
PFH of SpA/extra articular Features—N (%)	0 (0)	86
AS—N (%)	52 (60.5)	
IBD SpA—N (%)	16 (18.6)	
PsA—N (%)	8 (9.3)	
ReA—N (%)	4 (4.7)	
uSpA—N (%)	6 (6.9)	
SAPHO—N (%)	0 (0)	
Symptoms at disease onset		
Peripheral features—N (%)	41 (46.7)	86
Axial features—N (%)	23 (26.7)	86
Mixed axial and peripheral features—N (%)	22 (25.6)	86
Axial symptoms at follow up		
Inflammatory back pain—N (%)	55 (63.9)	86
Buttock pain—N (%)	25 (29.1)	86
Spinal stiffness—N (%)	30 (34.9)	86
Extra articular features		
IBD—N (%)	20 (23.3)	86
Psoriasis—N (%)	8 (9.3)	86
Acute Anterior Uveitis (AAU)—N (%)	9 (10.5)	86
HLA-B27 positive—N (%)	30 (42.2)	71
HIV positive—N (%)	0 (0)	86
Raised CRP > 10 mg/L	48 (55.8)	86
Sacro iliitis at diagnosis—N (%)	42 (65.6)	64 *
Radiographic sacro iliitis	40 (62.5)	64
Magnetic Sacro iliitis	2 (3.1)	64

AS: ankylosing spondylitis, IBD SpA: inflammatory-bowel-disease-associated spondyloarthritis, PsA: psoriatic arthritis, ReA: reactive arthritis, uSpA: undifferentiated spondyloarthritis, SAPHO: synovitis, acne, pustulosis. * 64 patients had sacroiliac X-rays available.

**Table 2 jcm-11-01299-t002:** Characteristics of Martinican SpA subtypes.

	AS *n* = 52	uSpA *n* = 6	ReA *n* = 4	PsA *n* = 8	IBD SpA *n* = 16	Total Patients Analyzed (N)
Age at diagnosis—mean ± SD (min–max)	42 ± 15.1 (22–67)	34 ± 13.5 (25–47)	39 ± 15.5 (18–55)	54 ± 14.2 (35–79)	36.0 ± 10 (18–54)	86
Male—N (%)	31 (59.6)	3 (50)	4 (100)	5 (62.5)	5 (31.3)	86
Inflammatory Back pain at follow up—N (%)	38 (73.1) *	2 (33.3)	0 (0)	2 (25)	13 (81.3)	86
Spinal stiffness—N (%)	26 (50)	0(0)	0(0)	1 (12.5)	4 (25)	86
Peripheral features—N (%)	33 (63.5)	4 (66.7)	4 (100)	8 (100)	14 (87.5)	86
IBD—N (%)	4 (7.6)		0 (0)	0 (0)	16 (100)	86
AAU—N (%)	8 (15.4)	1 (16.7)	0 (0)	0 (0)	0 (0)	86
Sacroiliitis—N (%)	34/41 (82.9) *	1/4 (25)	0/0 (0)	1/5 (20)	6/14 (42.9)	64
HLAB27 positive—N (%)	25/46 (54.3) *	0/5 (0)	2/3 (66.7)	1/8 (12.5)	2/9 (18.2)	71

AS: ankylosing spondylitis, IBD SpA: inflammatory-bowel-disease-associated spondyloarthritis, PsA: psoriatic arthritis, ReA: reactive arthritis, uSpA: undifferentiated spondyloarthritis. * Statistical association with AS when compared to overall SpA subtypes *p* = 0.029, *p* < 0.001 and *p* = 0.0027 for back pain, sacroiliitis and HLAB27, respectively.

**Table 3 jcm-11-01299-t003:** Therapeutics used in Martinican SpA subtypes.

Treatments	AS *n* = 52	uSpA *n* = 6	ReA *n* = 4	PsA *n* = 8	IBD SpA *n* = 16	All SpA *n* = 86
NSAIDS—N (%)	32 (61.5)	2 (33.3)	3 (75)	5 (62.5)	5 (31.3)	47 (54.65)
Steroids—N (%)	17 (32.7)	1 (16.7)	1 (25)	5 (62.5)	10 (62.5)	34 (39.5)
Methotrexate—N (%)	28 (53.8)	1 (16.7)	1 (25)	8 (100)	10 (62.5)	48 (55.8)
Sulfasalazine—N (%)	18 (34.6)	1 (16.7)	2 (50)	1 (12.5)	4 (25)	26 (30.2)
Leflunomide—N (%)	1 (1.9)	0 (0)	0 (0)	0 (0)	0 (0)	1 (1.2)
Pamidronate—N (%)	3 (5.7)	0 (0)	0 (0)	0 (0)	0 (0)	3 (3.5)
TNF inhibitors—N (%)	25 (48.1)	0 (0)	0 (0)	4 (50)	11 (68.8)	40 (46.5)

AS: ankylosing spondylitis, IBD SpA: inflammatory-bowel-disease-associated spondyloarthritis, PsA: psoriatic arthritis, ReA: reactive arthritis, uSpA: undifferentiated spondyloarthritis, NSAIDs: non-steroidal anti-inflammatory drugs.

## Data Availability

Data supporting reported results can be supplied from Fabienne Louis-Sidney, the University Hospital of Fort de France, Martinique.

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
