# Peer review of "Epidemiology and Characteristics of Spondyloarthritis in the Predominantly Afro-Descendant Population of Martinique, a French Caribbean Island"

_jcm, 2022, doi:10.3390/jcm11051299_

Round 1
Reviewer 1 Report
The paper is well written and the characteristics of SpA in an afro-descendent population is certainly a topic of interest to me. Unfortunately, the method chosen to conduct the study is prone to bias and therefore the reader doesn't know if the findings are reliable or due to the particular sub group of the population studied and the way the data was collected.
The authors themselves have hinted at some of the problems. Relying on hospital records to carry out a detailed review of a condition is fraught with problems due to inaccurate classification and missing data. The authors have shown information about wrong diagnostic codes in 44 of 146 patients identified, but they did not carry out a protocol guided clinical evaluation of the 86 patients with 'confirmed SpA diagnosis' and then highlight further errors and missing data. Not knowing how many had missing HLA B27 tests/Xrays/MRIs is particularly problematic. Although a family history may not have been mentioned in the hospital record it does not mean that this information has been sought and recorded for every patient.
Another major source of bias in epidemiological population studies is choosing to base the study in a university hospital rather than in the community. As alluded to in the discussion, the high prevalence of IBD could be a characteristic of the SpA population in Martinique OR it could simply be a sampling error due to the hospital providing a good secondary care clinic for IBD whilst neighboring private clinics might provide an alternative place of care for patients with psoriatic arthritis or ankylosing spondylitis.
Unfortunately these problems will not be easy to address. The minimum required would be to carry out a second community based study & then carry out a careful clinical review of all the community and hospital patients minimizing any gaps in the key diagnostic information.
Author Response
- The paper is well written and the characteristics of SpA in an afro- descendent population is certainly a topic of interest to me. Unfortunately, the method chosen to conduct the study is prone to group of the population studied and the way the data was collected. The authors themselves have hinted at some of the problems. Relying on hospital records to carry out a detailed review of a condition is fraught with problems due to inaccurate classification and missing data. The authors have shown information about wrong diagnostic codes in 44 of 146 patients identified, but they did not carry out a protocol guided clinical evaluation of the 86 patients with 'confirmed SpA diagnosis' and then highlight further errors and missing data.
Collection of SpA patients was done retrospectively through the hospital database.
However, patients were seen by the rheumatology and Internal medicine teams from the same hospital and patients’ files (from which the data were extracted) were complete enough to recover sufficient diagnosis criteria.
- Not knowing how many had missing HLA B27 tests/Xrays/MRIs is particularly problematic. Although a family history may not have been mentioned in the hospital record it does not mean that this information has been sought and recorded for every patient.
Missing datas are mentioned in Table 1 and Table 2.
Family history was systematically reported in every patient’s files.
- Another major source of bias in epidemiological population studies is choosing to base the study in a university hospital rather than in the community.
The particularity of Martinique is the existence of a unique hospital on a small island and that it is a university hospital. The internists practice exclusively in this hospital. The rheumatologists who practice outside of the hospital are not in private clinics and refer all their complex patients or those on biologics to the hospital rheumatologists. There are no other internal medicine or rheumatology services outside the university hospital. Relationships are tight between rheumatologists practicing in the hospital and those practicing outside of it.
- As alluded to in the discussion, the high prevalence of IBD could be a characteristic of the SpA population in Martinique OR it could simply be a sampling error due to the hospital providing a good secondary care clinic for IBD whilst neighboring private clinics might provide an alternative place of care for patients with psoriatic arthritis or ankylosing spondylitis.
No private clinic provides care for SpA patients in Martinique as mentioned earlier.
- Unfortunately these problems will not be easy to address. The minimum required would be to carry out a second community based study & then carry out a careful clinical review of all the community and hospital patients minimizing any gaps in the key diagnostic information.
Reviewer 2 Report
This is an interesting paper, which reports the clinical characteristics of a Martinican spondyloarthritis population. The authors included a small number of patients with a majority of ankylosing spondylitis. HLA-B27 was found in 42% of cases. The most frequent extra-articular manifestations found were IBD.
Major points:
- The authors use the term "Afro-descendant population". However, as the authors note in the limitations of this study (Line 192), the ethnicity of the patients included in this study could not be determined. Historically, the majority of the Martinican population is descended from Central and West Africa, but there are no details of the study population and the authors do not report the proportion of patients of Caucasian origin in the study population. It therefore seems more accurate not to use the term "Afro-descendant population" but the term "Martinican patient".
Title: we will change the title “Spondyloarthritis in afro decendant population” to “Spondyloarthritis in Martinican patient”.
What does “stage 1” mean?
Introduction:
- The authors mention a low representativeness of HLA-B27 in their population and rely on references 12 and 14: unless we are mistaken, the ref 14 does not provide any information about HLA-B27. We did not succeed to acces to ref 12. please check and correct the references if it's not appropriate.
Material & methods :
- Line 53: please detail "insufficient clinical and para-clinical information »
- Line 56: Please provide details of ICD-10 codes
- Line 59: please check this sentence. "According to point 23 of this declaration, an approval from the institutional review board (IRB) of University Hospital of Fort de France".
- Line 75: Are these treatments ongoing at the time of inclusion or these are treatments received since the onset of the disease?
- Line 77: Were there any biological treatments other than anti-TNF-a (Anti IL-17, anti IL-12/23, anti IL-23)?
- Statistical analysis:
- for subtypes analysis, We think non parametric test were more appropriate when number of patients is lower than 30. Also, the description of quantitative variables (median (quartiles)) should be reviewed in these groups < 30 patients.
Results :
- line 91: "86 patients met diagnostic criteria for SpA": There are no diagnostic criteria for SpA (except NY criteria). Are the authors talking about classification criteria? (if so which ones?): please rephrase this sentence.
- Table 1: How was the inflammatory syndrome defined? Do the authors have details of the inflammatory syndrome (mean CRP and/or ESR)?
- Table 2:
- Group uSpA, ReA, PsA, IBD: use non-parametric test
- Sacro iliitis line: there must be an error: 41 + 4 + 5 + 24 = 74. How can there be 6/24 sacro iliitis in the IBD group (N = 16)?
- Probable error => 6/14 and not 6/24
- Please provide a caption for the abbreviations used.
- Line 107: "p=0.029, p<10-3 and p=2,7.10-3 (Table 2)”. This would be clearer if the authors used "p = 0.029, p < 0.001 and p = 0.0027".
Discussion:
- Line 115: it would be clearer, if the authors used the prevalence rate/100000 inhabitants in Martinique and elsewhere in the world (cf ref: 1093/rheumatology/ket387) (same presentation as the rest of the world SpA prevalence).
- How do the authors explain the higher age of diagnosis in their population compared to the cohorts cited?
- How do the authors explain the absence of familial history when 40% of the patients in the cohort are HLA-B27 positive?
- Line 139: "In 2001, Bera et al. reported the absence of HLAB27 allele in the Martinican population,"
- This sentence is not entirely true; Bera et al find a distribution of 0.02 in the Martinique population with no difference from the French and Congolese populations (see table 2 of Bera et al).
Minor points:
- Please correct "HLAB27" to "HLA-B27
- Correct the bibliographic references according to the Mdpi style with the numbering of the references.
Author Response
Major points:
- The authors use the term "Afro-descendant population". However, as the authors note in the limitations of this study (Line 192), the ethnicity of the patients included in this study could not be determined. Historically, the majority of the Martinican population is descended from Central and West Africa, but there are no details of the study population and the authors do not report the proportion of patients of Caucasian origin in the study population. It therefore seems more accurate not to use the term "Afro-descendant population" but the term "Martinican patient".
Title: change the title “Spondyloarthritis in afro decendant population” to “Spondyloarthritis in Martinican patient”.
Ok , we will change the title.
What does “stage 1” mean?
We were asked to do so in the instructions for authors: The title of the paper should include words “Stage 1”.
But we will take it off.
Introduction:
- The authors mention a low representativeness of HLA-B27 in their population and rely on references 12 and 14: unless we are mistaken, the ref 14 does not provide any information about HLA-B27. We did not succeed to acces to ref 12. please check and correct the references if it's not appropriate.
External identifier for reference 12 is urn:oclc:record:1097262803. We will add it in the reference.
References 12 and 14 provide information about ethnicity of the Martinican population.
Reference 13 provides informations for HLA-B27 low representativeness.
References 12-14 was including ref 12,13,14.
We will do corrections to clarify.
Material & methods :
- Line 53: please detail "insufficient clinical and para-clinical information »
Insufficient clinical and para clinical information in patients’ medical file to make a diagnosis of spondyloarthritis. Clinical information missing was back pain, arthritis, enthesitis, dactylitis, buttock pain, spinal stiffness, presence of clinical features specific to another inflammatory rheumatism.
Missing paraclinical information was either Sacro iliac and spinal x rays, HLA-B27 or immunological test for another possible inflammatory rheumatism.
- Line 56: Please provide details of ICD-10 codes
Ok
- M02(Postinfective and reactive arthropathies),
- M03 (Postmeningococcal arthritis),
- M45 (Ankylosing spondylitis),
- M07(Enteropathic arthropathies),
- M08 (Juvenile arthritis),
- M09 (Juvenile arthritis in diseases classified elsewhere),
- 9 (Cervical disc disorder, unspecified)
- 4(Arthropathy in Crohn's disease)
- 5( rthropathy in ulcerative colitis)
- K51 (ulcerative colitis)
- 5 (Arthropathic psoriasis): we missed this one
- Line 59: please check this sentence. "According to point 23 of this declaration, an approval from the institutional review board (IRB) of University Hospital of Fort de France".
Ok
- Line 75: Are these treatments ongoing at the time of inclusion or these are treatments received since the onset of the disease?
These treatments were received since the onset of the disease.
We will add it in the text
- Line 77: Were there any biological treatments other than anti-TNF-a (Anti IL-17, anti IL-12/23, anti IL-23)?
No there wasn’t. We wanted to to described Spondyloarthritis in the Martinican populations, which is mostly of African descent, before the ASAS criteria publication in 2009. So, we did a retrospective study between 1997 and 2008. IL-17, IL-23 and IL-12/23 were not used at that time.
- Statistical analysis:
- for subtypes analysis, We think non parametric test were more appropriate when number of patients is lower than 30. Also, the description of quantitative variables (median (quartiles)) should be reviewed in these groups < 30 patients.
We did a subtypes analysis between Anlylosing spondilitys ( 52 patients ) and overall subtypes of spondylarthritis ( 34 patients).
For description of quantitative variables, the person in charge of the database and statistical analysis is absent.
We can inform on the minimal and maximal of age for the study population and for each group. We will put them in Table 1 and Table 2
Spondyloarthritis: min 18 years old / max 79 years old
Ankylosing spondylitis: min 22 years old / max 67 years old
Psoriatic arthritis : min 35 years old / max 55 years old
Reactive arthritis : : min 18 years old / max 55 years old
Undifferentiated arthritis: : min 25 years old / max 47 years old
Inflammatory bowel disease associated spondyloarthritis : : min 18 years old / max 54 years old
Results :
- line 91: "86 patients met diagnostic criteria for SpA": There are no diagnostic criteria for SpA (except NY criteria). Are the authors talking about classification criteria? (if so which ones?): please rephrase this sentence.
86 patients met the “European spondylarthropathy study group” (ESSG) criteria for SpA. uSpA was diagnosed if the ESSG criteria werefulfilled, but no diagnosis of AS, PsA, ReA or IBD associated SpA.
- Table 1: How was the inflammatory syndrome defined? Do the authors have details of the inflammatory syndrome (mean CRP and/or ESR)?
As mentionned in the Methods, inflammatory syndrome was defined as inflammatory syndrome at diagnosis defined by a C reactive protein > 10mg/l.
We will mentioned it also in table 1.
- Table 2:
- Group uSpA, ReA, PsA, IBD: use non-parametric test
We did a subtypes analysis between Anlylosing spondilitys (52 patients) and overall subtypes of spondylarthritis ( 34 patients).
- Sacro iliitis line: there must be an error: 41 + 4 + 5 + 24 = 74. How can there be 6/24 sacro iliitis in the IBD group (N = 16)?
- Probable error => 6/14 and not 6/24
There was an error : it was 6/14 and not 6/24
- Please provide a caption for the abbreviations used.
Ok
- Line 107: "p=0.029, p<10-3 and p=2,7.10-3 (Table 2)”. This would be clearer if the authors used "p = 0.029, p < 0.001 and p = 0.0027".
Ok
Discussion:
- Line 115: it would be clearer, if the authors used the prevalence rate/100000 inhabitants in Martinique and elsewhere in the world (cf ref: 10.93/rheumatology/ket387) (same presentation as the rest of the world SpA prevalence)
ok
- How do the authors explain the higher age of diagnosis in their population compared to the cohorts cited?
We could not determinate the proportion of African descendant in the study population. In Bera et al. study, African characteristic HLA alleles were significantly represented in Martinicans which reflects the African genetic background of this population. Genetic factors certainly play a role in the disease’s phenotype and expression. This needs to be investigated.
- How do the authors explain the absence of familial history when 40% of the patients in the cohort are HLA-B27 positive?
HLA-B27 has been reported to be associated with personal family history (PFH) but we also see in African American, Afro Caribbean in England and in African SpA cohorts a paucity even absence of PFH. Caribbean populations have the particularity of being part of the African diaspora. We know that HLA-B27 is associated with higher risk of developing SpA but there might also be in African populations some protective factors. Moreover, others non-HLA polymorphisms are implicated in the disease phenotype and need to be investigated.
- Line 139: "In 2001, Bera et al. reported the absence of HLAB27 allele in the Martinican population,"
-
- This sentence is not entirely true; Bera et al find a distribution of 0.02 in the Martinique population with no difference from the French and Congolese populations (see table 2 of Bera et al).
Ok , we’ll make the correction.
Minor points :
Please correct "HLAB27" to "HLA-B27
Ok
Correct the bibliographic references according to the Mdpi style
with the numbering of the references.
Ok
Reviewer 3 Report
General comments:
The authors aimed to describe the characteristics of Spondyloarthritis in the Martinican Afro-descendant population studying a retrospective hospital-based cohort of 86 patients followed in a high level of care setting.
Results highlight some differences in the clinical presentation of SPAs in these patients with respect to their European and North American homologues, particularly the lower prevalence of HLA B27, older age at diagnosis, the preponderance of peripheral symptoms and the lower prevalence of psoriatic arthritis. These findings are in agreement with other studies on patients of African ethnicity.
In the discussion, the authors correctly underline the main points of weakness of the study consisting specifically in its retrospective nature, the hospital-based setting and the missing radiological data on sacroiliitis.
The interplay between different genetic backgrounds and different environments may result in a different disease presentation and the study adds pieces of information on the spectrum of SpAs in African or their descendant patients.
Specific comments:
rows 42 and 43: maybe, citations 13 and 14 are to be inverted in the references
row 43 and row 158-160: please, specify the HLA-B27 prevalence in the Martinican population according to the study cited (Bera et al.)
row 58: please, cite the study with the ESSG criteria (Dougados et al. 1991)
row 82: please, specify the criteria adopted in the study for the definition of sacroiliitis at conventional radiology (X-rays) and magnetic resonance [correct "magnetic" in the text]
row 115: please, utilize the term “raised CRP ...” rather than “Inflammatory syndrome …”
Table 1: please,
- express age at diagnosis as mean value ±SD (min-max), and delete rows for minimal and maximal age at diagnosis;
- express the duration of disease as mean value ±SD (min-max);
- “Family History” and “Extra-Articular features” are two separate items and might be better placed in two rows;
- Diagnosis could be abbreviated as AS, IBD-SpA, PsA, ReA, uSpA and SAPHO;
- Moreover, “SAPHO” should be mentioned in the Methods section among the diagnosis and the acronym should be also described in the text of this section;
- Delete the row “Biology” and maintain in separate rows the items “HLA-B27”, “HIV positive” and “CRP > 10 mg/L” [delete the descriptor “inflammatory syndrome” and the relative footnote];
- All abbreviations and acronyms could be described in the footnotes of the table;
- The total number (64) of sacroiliitis probably refers to the total of the patients with positive X-rays or MRI (or both). It should be also specified how many X-rays and how many MRIs were globally available for evaluation in the total number (86) of patients. Relative percentages (positive/total) should be specified for each imaging method accordingly.
Table 2: please,
- express age at diagnosis as mean value ±SD (min-max), and delete rows for minimal and maximal age at diagnosis;
- All abbreviations and acronyms could be described in the footnotes of the table.
Author Response
Specific comments:
rows 42 and 43: maybe, citations 13 and 14 are to be inverted in the references
Thank you, We’ve made the correction.
row 43 and row 158-160: please, specify the HLA-B27 prevalence in the Martinican population according to the study cited (Bera et al.)
We’ve specified HLA-B27 prevalence as requested.
row 58: please, cite the study with the ESSG criteria (Dougados et al. 1991)
Thank you, We’ve made the correction
row 82: please, specify the criteria adopted in the study for the definition of sacroiliitis at conventional radiology (X-rays) and magnetic resonance [correct "magnetic" in the text]
Thank you, We’ve made the correction
row 115: please, utilize the term “raised CRP ...” rather than “Inflammatory syndrome …”
Thank you, We’ve made the correction
Table 1: please,
- express age at diagnosis as mean value ±SD (min-max), and delete rows for minimal and maximal age at diagnosis; Thank you, We’ve made the correction
- express the duration of disease as mean value ±SD (min-max); Thank you, We’ve made the correction
- “Family History” and “Extra-Articular features” are two separate items and might be better placed in two rows.
In the row family history of SpA or Extra articular features we meant family history of SpA or family history of Extra articular features of SpA such as Anterior uveitis , psoriasis or IBD . We’ve put it like that in the row : “PFH of SpA/extra-articular features “. Is it convenient for you ?
- Diagnosis could be abbreviated as AS, IBD-SpA, PsA, ReA, uSpA and SAPHO; Thank you, We’ve made the correction
- Moreover, “SAPHO” should be mentioned in the Methods section among the diagnosis and the acronym should be also described in the text of this section; Thank you, we mentioned it in the methods as requested
- Delete the row “Biology” and maintain in separate rows the items “HLA-B27”, “HIV positive” and “CRP > 10 mg/L” [delete the descriptor “inflammatory syndrome” and the relative footnote]; Thank you, We’ve made the correction
- All abbreviations and acronyms could be described in the footnotes of the table
Thank you, We’ve made the correction
- The total number (64) of sacroiliitis probably refers to the total of the patients with positive X-rays or MRI (or both). It should be also specified how many X-rays and how many MRIs were globally available for evaluation in the total number (86) of patients. Relative percentages (positive/total) should be specified for each imaging method accordingly.
At the following time of this cohort, MRI criteria for sacro iliitis were not well defined. Moreover, at that time there was only one MRI for the entire island and sacro iliac magnetic resonance was not done in routine. Radiographic sacro-iliac images were always done first. We took them from the medical records and reviewed each according to the New York criteria of 1966. If normal, patients ‘files were reviewed to see whether or not an MRI positive result had been mentioned. 64 patients had radiographic sacro-iliac images disponible. Out of 24 patients with normal X rays, only two had MRI positive results mentioned in medical records. It is difficult to confirm the right number of MRI performed. However, because access to MRI was poor at that time, we can easily suppose that it wasn’t done for the 24 patients with normal radiographic sacro-iliac images.
Table 2: please,
- express age at diagnosis as mean value ±SD (min-max), and delete rows for minimal and maximal age at diagnosis. Thank you, We’ve made the correction
- All abbreviations and acronyms could be described in the footnotes of the table. Thank you, We’ve made the correction
Round 2
Reviewer 1 Report
I note the author's comments about the unique situation of the hospital and that private rheumatologists 'refer all their complex patients or those on biologics'. This sort of selection from primary to secondary care is precisely why epidemiological studies should use a community based population.
I also note the comment that the hospital records are complete and enable accurate diagnoses to be made. This might be true but medical records are not usually 100% accurate and if some patients have not had MRIs at baseline they may be an unrecognised group of AxSpA patients who have been reviewed but not specifically labelled as having AS.
Author Response
I note the author's comments about the unique situation of the hospital and that private rheumatologists 'refer all their complex patients or those on biologics'. This sort of selection from primary to secondary care is precisely why epidemiological studies should use a community based population.
I also note the comment that the hospital records are complete and enable accurate diagnoses to be made. This might be true but medical records are not usually 100% accurate and if some patients have not had MRIs at baseline they may be an unrecognised group of AxSpA patients who have been reviewed but not specifically labelled as having AS.
You’re right. We’ve put it in the discussion/ conclusion and we’re currently investigating evolution of SpA epidemiology in Martinique starting from 2009, when ASAS criteria for AxSpA were proposed.
Reviewer 2 Report
Comments :
Title : try to emphasize the title; for example : "spondyloarthritis characteristics in the martinican population”
In the list of authors, change the place of * in front of the corresponding author
As mentioned in the first comments, it seems necessary to replace the term "Afro-descendant population" by "Martinican population", as the authors cannot state the proportion of African-descendants in the studied population.
- Line 12, 21, 46, 132, 218
Abstract :
Background : please modify taking into account these comments (line 10, line 12 (we aim to assess characteristics of Spa in martinican population.... )
Line 20-21 : this is the first descriptive .... in martinican patients.
Key words: remove ethnicity, African, Afro-descendant. Put in the term "martinican".
Introduction :
- I will start the introduction by talking about the absence of data concerning spondyloarthropathies in the Martinican population, then evoke the fact that the majority of the patients are African descendant and go on to discuss the characteristics of SpA in this population;
- "These features are incorporated in several SpA classification criteria and particularly in the most recent 2009 ASAS criteria [7]. "
- include the ASAS criteria reference: http://dx.doi.org/10.1136/ard.2009.108233
- Line 46: please remove "afro-descendant population" and replace by, for example, "we aim to assess characteritics of SpA patients in a population of a high level of care in Martinique"
Material and methods:
- I don't understand how you classified the patients:
- The ESSG criteria allow the diagnosis of spondyloarthritis but how did you classify patients into Ankylosing Spondylitis (New York criteria?), Psoriatic Arthritis (CASPAR)...
- Please provide details in the text
- Line 52: include reference to ESSG criteria (DOI: 1136/ard.2010.133645)
Results:
- Delete the minimal and maximal age in table 1 & 2 as it is not clearly presented in this way.
Conclusion:
- Line 217: I would replace 'caribbean' by 'martinican patient' as 'caribbean' is too broad.
- Please correct reference 46 (quote the original article and not the correction): http://dx.doi.org/10.1136/ard.2009.108233
- Ref 46 : Correction: The development of assessment of SpondyloArthritis international society classification criteria for axial spondyloarthritis (part II): validation and final selection. Ann Rheum Dis 2019;78:e59.2-e59. https://doi.org/10.1136/ard.2009.108233corr1.
Author Response
Title : try to emphasize the title; for example : "spondyloarthritis characteristics in the martinican population”
We propose :“Epidemiology and characteristics of Spondyloarthritis in the predominantly afro descendant population of Martinique, a French Caribbean island”
We mentioned in the title the unique characteristic of Martinique which population is mostly of African descent without making any presumption about ethnicity of our SpA cohort. Is this convenient for you?
In the list of authors, change the place of * in front of the corresponding author
Thank you, we've made the correction
As mentioned in the first comments, it seems necessary to replace the term "Afro-descendant population" by "Martinican population", as the authors cannot state the proportion of African-descendants in the studied population.
- Line 12, 21, 46, 132, 218
Thank you. We’ve made the corrections.
For line 218 the afro descendant population we’re talking about the one from the English PsA cohort where the term afro-Caribbean is used since ethnic studies are allowed there. We put some precisions in the discussion to avoid misunderstanding. Is this convenient for you?
Abstract :
Background : please modify taking into account these comments (line 10, line 12 (we aim to assess characteristics of Spa in martinican population.... ) Thank you , correction has been made.
Line 20-21 : this is the first descriptive .... in martinican patients. Thank you , correction has been made.
Key words: remove ethnicity, African, Afro-descendant. Put in the term "martinican".
Corrections has been made. We’ve put Martinique instead of martican as Martinique is recognized as a MeSH and not Martinican. Is that ok for you ?
Introduction :
- I will start the introduction by talking about the absence of data concerning spondyloarthropathies in the Martinican population, then evoke the fact that the majority of the patients are African descendant and go on to discuss the characteristics of SpA in this population.
Thank you , We’ve made correction as requested.
- "These features are incorporated in several SpA classification criteria and particularly in the most recent 2009 ASAS criteria [7]. "
- include the ASAS criteria reference: http://dx.doi.org/10.1136/ard.2009.108233
Thank you for the reference. We’ve made corrections as requested.
- Line 46: please remove "afro-descendant population" and replace by, for example, "we aim to assess characteritics of SpA patients in a population of a high level of care in Martinique"
We’ve made corrections as requested.
Material and methods:
- I don't understand how you classified the patients:
- The ESSG criteria allow the diagnosis of spondyloarthritis but how did you classify patients into Ankylosing Spondylitis (New York criteria?), Psoriatic Arthritis (CASPAR)...
- Please provide details in the text
We’re sorry for the misunderstanding. We made some corrections and provided details as requested.
- Ankylosing spondylitis was defined by modified New York criteria. We had said it in the result but didn’t mention it in the methods. Correction was made as requested.
Psoriatic arthritis was diagnosed based on CASPAR criteria. - Undifferenciated SpA is defined in the methods
- We put SAPHO criteria also
- All criteria is supported with a reference.
Is it convenient for you?
- Line 52: include reference to ESSG criteria (DOI: 1136/ard.2010.133645)
We’ve included reference for ESSG criteria (Dougados et al. 1991 / DOI : 10.1002/art.1780341003).
We couldn’t find yours, is that one convenient ?
Results:
- Delete the minimal and maximal age in table 1 & 2 as it is not clearly presented in this way.
We’ve made the correction.
Conclusion:
- Line 217: I would replace 'caribbean' by 'martinican patient' as 'caribbean' is too broad.
We’ve made the correction.
Please correct reference 46 (quote the original article and not the correction): http://dx.doi.org/10.1136/ard.2009.108233
Thank you, We’ve made the correction.
- Ref 46 : Correction: The development of assessment of SpondyloArthritis international society classification criteria for axial spondyloarthritis (part II): validation and final selection. Ann Rheum Dis 2019;78:e59.2-e59. https://doi.org/10.1136/ard.2009.108233corr1.
Thank you, We’ve made the correction.